# Production of an Active, Human Membrane Protein in *Saccharomyces cerevisiae*: Full-Length FICD

**DOI:** 10.3390/ijms23052458

**Published:** 2022-02-23

**Authors:** Minttu S. Virolainen, Cecilie L. Søltoft, Per A. Pedersen, Lars Ellgaard

**Affiliations:** 1Linderstrøm-Lang Centre for Protein Science, Department of Biology, University of Copenhagen, Ole Maaløes Vej 5, DK-2200 Copenhagen, Denmark; minttu.virolainen@bio.ku.dk (M.S.V.); cecilie.soltoft@bio.ku.dk (C.L.S.); 2Department of Biology, University of Copenhagen, Universitetsparken 15, DK-2200 Copenhagen, Denmark

**Keywords:** AMPylation, Fic proteins, FICD, membrane protein purification, *Saccharomyces cerevisiae*, recombinant protein expression

## Abstract

The human Fic domain-containing protein (FICD) is a type II endoplasmic reticulum (ER) membrane protein that is important for the maintenance of ER proteostasis. Structural and in vitro biochemical characterisation of FICD AMPylase and deAMPylase activity have been restricted to the soluble ER-luminal domain produced in *Escherichia coli*. Information about potentially important features, such as structural motifs, modulator binding sites or other regulatory elements, is therefore missing for the approximately 100 N-terminal residues including the transmembrane region of FICD. Expressing and purifying the required quantity and quality of membrane proteins is demanding because of the low yields and poor stability often observed. Here, we produce full-length FICD by combining a *Saccharomyces cerevisiae*-based platform with green fluorescent protein (GFP) tagging to optimise the conditions for expression, solubilisation and purification. We subsequently employ these conditions to purify milligram quantities of His-tagged FICD per litre of culture, and show that the purified, detergent-solubilised membrane protein is an active deAMPylating enzyme. Our work provides a straightforward methodology for producing not only full-length FICD, but also other membrane proteins in *S. cerevisiae* for structural and biochemical characterisation.

## 1. Introduction

Fic (filamentation induced by cAMP) proteins are ubiquitous, structurally conserved enzymes that post-translationally modify hydroxyl-containing residues by cleaving high-energy phosphate bonds on small co-substrates, such as ATP, and transferring phosphate-containing groups to substrates [1]. Fic proteins are prevalent in bacteria, and only a single *fic* gene has been identified in metazoans [2,3]. The human Fic domain-containing protein, FICD (Figure 1A), is a type II transmembrane protein that localises to the endoplasmic reticulum (ER), and to a lesser extent to the nuclear envelope [4,5,6,7,8].

FICD catalyses two antagonistic reactions by utilising an oligomerisation-induced switch in the structure of the active site; AMPylation by the monomeric enzyme and deAMPylation by the dimeric enzyme [4,9,10,11,12]. These FICD-mediated reactions are important for the maintenance of ER proteostasis [4,5,13,14] and neuronal function [6,15,16,17,18,19], as well as adaptive immunity [19]. The best-known FICD substrate is BiP, a central Hsp70-family chaperone, which catalyses ER protein folding and regulates the unfolded protein response [4,5,20]. FICD AMPylates BiP at its substrate-binding domain, causing inactivation [4,5,13,21]. Conversely, deAMPylation by FICD reactivates BiP [9,10]. Overall, present findings indicate an important role for FICD in modulating the ER protein folding capacity in response to alterations in the ER protein folding load [4,9,13].

Membrane proteins constitute 25–30% of both the eukaryotic and prokaryotic proteomes [24,25,26], but are strongly underrepresented in the protein data bank [27,28]. Moreover, while single-pass membrane proteins such as FICD represent roughly half of all membrane proteins, structural studies of these are limited [24,29,30]. The currently available crystal structures of FICD include the catalytic domain and the adjacent tetratricopeptide repeats (TPRs) (Figure 1B) [10,11,12,22]. In vitro investigations of FICD’s structure and enzymatic mechanism have been conducted using N-terminally truncated (lacking the first approximately 100 residues), soluble protein variants produced in *E. coli*. Thus, the transmembrane domain and the membrane-adjacent regions (residues 1–102) have not been functionally or structurally characterised although they may harbour important determinants regulating, e.g., activity, localisation and protein–protein interactions.

Obtaining sufficient quantities of pure membrane protein can be both challenging and time consuming due to low expression levels, solubility issues and challenges in obtaining and maintaining uniformly folded protein. To overcome some of these challenges, we employed the previously developed PAP1500 *S. cerevisiae*-based membrane protein expression platform [31,32,33,34,35,36], and coupled this with GFP tagging to produce full-length human FICD (Figure 2). This platform has been successfully used to produce multi-spanning membrane proteins such as aquaporins, transmembrane ATPases and ion channels [32,33,34,35,37,38,39,40]. The approach bypasses traditional cloning procedures that can be time consuming and costly, and lead to the inclusion of cloning scars [41]. Instead, the platform utilises the ability of *S. cerevisiae* to recombine linear DNA fragments with overlapping segments, such as PCR products, into a mature expression vector [42,43]. The host strain overproduces the Gal4 transcriptional activator, which works in concert with the strong galactose-inducible CYC-GAL promoter in the pEMBLyex4 shuttle vector (Figure 3A). This promoter combines the constitutive cytochrome c1 (CYC1) promoter with the upstream activating sequence from the GAL1-GAL10 promoter [44]. This expression system ensures that the plasmid copy number can be increased from about 20 to 200–400 copies per cell prior to the induction of expression by starving the cells for leucine [37,45]. This platform has been shown to lead to exceptionally high amounts of pure membrane protein (up to 8% of the total membrane protein content in the yeast cell [32]), compatible with biophysical and functional characterisation, including X-ray crystallography [32,34,37,46,47], cryo-EM [48] and SAXS [49].

To identify the optimal conditions for obtaining solubilised, correctly folded FICD, we introduced a GFP tag at its cytosolic N-terminal tail [47]. This allowed us to use small-scale expression tests to quickly optimise the expression conditions and identify suitable detergents for solubilisation, before initiating large-scale expression and purification of FICD without the GFP tag. A single round of expression produced approximately 20 mg of FICD in a buffer that is compatible with enzymatic and biophysical characterisation, as supported by the demonstration of deAMPylation activity by the purified protein.

## 2. Results

### 2.1. Recombinant GFP-FICD Is Expressed to a High Density in S. cerevisiae

To generate the expression vectors encoding human His_8_-GFP-TEV-FICD (GFP-FICD), including a TEV protease cleavage site, or His_8_-FICD (His-FICD), we utilised the ability of *S. cerevisiae* to recombine linear DNA fragments by homologous recombination (Figure 3A, Appendix A). We chose to perform FICD expression at 15 °C, as previous experiments have shown that although *S. cerevisiae* grows optimally at 30 °C, membrane protein folding and maturation is often compromised at this temperature [32,34]. To determine the appropriate induction time, the total cell GFP fluorescence was measured at 24 h intervals (Figure 3B). In total, 120 h after induction, fluorescence accumulation slowed down significantly, and the cells were harvested. GFP-FICD was expressed to a concentration of approximately 180 pmol/mg membrane protein, corresponding to approximately 1.5% of total membrane protein content. The optimal expression conditions obtained from the small-scale experiments were subsequently applied for large-scale production of the His-FICD construct in 15-litre bioreactors (see below).

### 2.2. GFP-FICD Likely Localises to the Golgi

In human cells, FICD localises to the ER membrane, where its C-terminal catalytic domain faces the ER lumen. We used fluorescence microscopy to ensure that heterologously expressed GFP-FICD was integrated into a membrane in our *S. cerevisiae* expression system. Here, the GFP signal did not display a pattern characteristic of ER localisation [50]. We compared the FICD signal to that of two GFP-tagged yeast membrane proteins of known localisation, the vacuolar Pmc1 Ca^2+^-ATPase and the Golgi-located Pmr1 Ca^2+^, Mn^2+^-ATPase [35,51,52] (Figure 3C). GFP-FICD showed a distribution similar to that of Pmr1-GFP, suggesting that in yeast this FICD fusion protein is most likely integrated into the Golgi membrane.

To investigate whether the GFP-FICD protein accumulated as a full-length protein, we separated the membrane proteins by SDS-PAGE and visualised the fusion protein using in-gel fluorescence (Figure 3D). The large majority of GFP-FICD migrated according to expectation at around 62–67 kDa, including a 10–15 kDa contribution from the correctly folded denaturation-resistant GFP tag [37,53]. A faint fluorescent band was visible at around 28 kDa, the molecular weight of GFP, indicating that the tag was removed from a very small fraction of the fusion protein.

### 2.3. Zwitterionic Detergents Recover More GFP-FICD Than Non-Ionic Detergents

We next carried out a screen to identify detergents (see Table 1) that could solubilise GFP-tagged FICD from the crude membranes, while preserving protein integrity (Figure 4). Detergents were either used alone or in combination with cholesteryl hemisuccinate (CHS), which may have a stabilising effect during membrane protein solubilisation and purification [33,35]. Most of the harsher ionic detergents were only tested in the presence of CHS. Overall the solubilisation efficiency varied substantially, ranging from about 5% to 100% solubility, with the zwitterionic Fos-cholines (FC12 and FC13) and n-dodecyl-dimethylamine oxide (LDAO) demonstrating higher recovery (70–100%) than the non-ionic detergents (Figure 4A,B). Fos-cholines recovered over 90% of the fluorescence when used in combination with CHS, whereas LDAO solubilised approximately 60% of GFP-FICD. For all zwitterionic detergent combinations, a 2:1 detergent to protein ratio yielded the best results.

Among the non-ionic detergents, the maltoside detergents DM, DDM and TDM solubilised an acceptable 30–40% of GFP-FICD, while HDM performed poorly (Figure 4A,B). The glucopyranoside detergents OG and NG only solubilised around 10% and 25% of the fusion protein, respectively. The maltopyranoside detergent MNG and the cyclohexyl maltoside detergent PCCM performed similarly to NG. The addition of CHS markedly lowered the yield in all detergent combinations excluding DM. The solubilisation efficiency of the non-ionic detergents could be improved slightly but consistently by increasing the detergent concentration (Figure 4A,B).

### 2.4. Cholesteryl Hemisuccinate Improves GFP-FICD Homogeneity

We next tested the best performing detergents for their ability to retain GFP-FICD in a folded conformation. For this purpose, we used FSEC, in which a monodisperse (symmetric) elution profile is indicative of folding and a unique oligomeric state [54,55,56,57]. The zwitterionic FC12 combined high solubilisation efficiency with high protein integrity, as shown by the monodisperse elution profile in the absence and presence of CHS (Figure 4C). GFP-FICD that was solubilised in the non-ionic DM or DDM eluted asymmetrically in the absence of CHS, whereas the addition of CHS led to a monodisperse elution profile (Figure 4D,E). In all conditions, only a small amount of fluorescent protein eluted in the void volume, likely representing higher molecular weight aggregates (Figure 4C–E). Although FC12 provided the highest overall yield, we decided to proceed with DM and DDM in conjunction with CHS, since these detergents are best suited for downstream applications such as biochemical characterisation and crystallography [58].

### 2.5. His-FICD Can Be Purified in a Folded Conformation

Structural and biochemical characterisation can be hampered by the presence of the large and bulky GFP tag. Since we were unable to efficiently remove the tag by TEV protease cleavage (Appendix A), we generated a His-tagged variant of our FICD construct, devoid of GFP. His-FICD was expressed and solubilised according to our optimised protocol, before subjecting the soluble fraction to immobilised metal affinity purification (IMAC). The identity of the purified protein was confirmed by SDS-PAGE and immuno -blotting (Figure 5A). The analysis revealed the presence of monomeric and oligomeric FICD species. Despite this apparent heterogeneity, during the subsequent size exclusion chromatography (SEC) analysis the protein eluted in a symmetric peak (Figure 5B), indicating that the His-tagged FICD adopts a folded, unique oligomeric state in solution.

The purified sample was further analysed by reducing and non-reducing SDS-PAGE (Figure 5C). A few high molecular weight species observed on the gel but not in the SEC analysis (Figure 5B) were partially sensitive to DTT, whereas the FICD dimer band was DTT resistant, suggesting that the purified protein might form non-covalent dimers. This would not be surprising, since N-terminally truncated FICD adopts a dimeric form in vitro [10,12], and non-covalent, SDS-resistant oligomerisation is often observed for oligomeric membrane proteins during SDS-PAGE analysis [37,59,60]. Overall, the chromatography analysis indicated that the GFP tag is not required for FICD stabilisation, and that the His-tagged variant remained folded upon purification.

### 2.6. Purified Full-Length His-FICD Is an Active deAMPylase

FICD’s switch from AMPylation to deAMPylation activity is governed by dimerisation at a low nanomolar concentration (K_D_ = 1.2 nM), as shown for the N-terminally truncated enzyme, which exists exclusively as a dimer in solution and possesses very weak AMPylation activity towards FICD’s natural substrate, BiP [9,10,12]. We therefore hypo-thesised that full-length His-FICD functions as a deAMPylase.

To test for deAMPylation activity, we first AMPylated BiP with radioactively labelled α-[^32^P]-ATP using the soluble L258D variant of FICD_Δ102_ (produced in *E. coli*), which is rendered monomeric by a mutation in the dimer interface and is known to efficiently AMPylate BIP [9,10,12,61]. AMPylated BiP (BiP-AMP) was then purified, and subsequently incubated in the presence or absence of His-FICD. The time-dependent removal of radiolabelled [^32^P]-AMP was quantified by phosphorimaging (Figure 5D,E). His-FICD readily removed ~80% of the AMP from BiP within 60 min with a half-time for deAMPylation of approximately 15 min, demonstrating that the enzyme functions as a deAMPylase in vitro. In the absence of the enzyme, the AMPylation level remained largely constant, indicating that the removal of AMP was due to the enzymatic activity of His-FICD, and not due to spontaneous hydrolysis. The minor increase in BiP AMPylation observed at the 2 min time-point is within the experimental error. Alternatively, it may be explained by the potential presence of residual amounts of ATP and monomeric FICD in the sample, carried over from the preceding AMPylation reaction. Regardless, full-length His-FICD deAMPylated BiP with comparable kinetics as observed for the N-terminally truncated enzyme purified from *E. coli* (see Discussion). Overall, the protocol presented here allowed the purification of full-length human FICD from yeast membranes in an active form, in a buffer that supports both structural and functional analysis.

## 3. Discussion

Most published crystal structures of membrane proteins are of prokaryotic origin [27]. Since approximately 85% of human membrane proteins do not have a prokaryotic ortholog, our knowledge of their structure and function is limited [62]. One such protein is the human ER membrane protein FICD, which has been studied in vitro using only its truncated, soluble form. Here, we have developed a protocol to express, solubilise, and purify functional, full-length human FICD using the *S. cerevisiae*-based PAP1500 membrane protein production platform [31,32,33,34,35]. Similar to bacteria, yeasts are inexpensive to cultivate and require relatively simple culture conditions. As a eukaryote, yeasts are more suitable for the expression of human proteins that require specific post-translational modifications for function or maturation. For our purposes, *S. cerevisiae* was an even more attractive host because yeast does not carry any known *fic* genes, preventing the inadvertent purification of heterodimers of yeast and human FICD. To streamline the expression, solubilisation and purification conditions, we coupled the expression platform with GFP tagging of FICD [47].

In human cells, FICD localises to the ER membrane in a type II orientation [4,12,17]. However, as FICD does not contain an arginine-based ER-retention motif [63,64], it is presently unclear how the protein is retained in the ER in human cells. In our *S. cerevisiae* expression system, the GFP-tagged FICD construct most likely localised to the Golgi membrane (Figure 3C). We and others have previously reported that some heterologously expressed proteins can be mislocalised in yeast [37,65], potentially caused by inherent differences in secretory system function or due to the strong expression of the proteins. It is also possible that the GFP tagging overrides or interferes with an ER retention signal. Despite the unexpected subcellular localisation, the fusion protein was not degraded. By quantifying the GFP fluorescence, we could estimate that the exogenous GFP-FICD was expressed to a density of ~181 pmol/mg membrane protein in crude membranes, over a 120 h expression period. This represents about 1.5% of the total membrane protein content (Figure 3B), which is sufficient for biochemical characterisation and structure determination.

GFP tagging ensured that we could easily test the ability of various detergents to solubilise and retain FICD in the folded conformation (as judged from FSEC elution profiles) [47,66]. By screening various detergents, FICD solubility was shown to be far greater in the relatively harsher zwitterionic detergents than in the milder non-ionic detergents (Figure 4A,B). The non-ionic DM and DDM also retained the protein in a folded conformation, and are generally more compatible with crystallography [58]. Due to its ten-fold lower critical micelle concentration, DDM is generally a more economical choice than either DM or FC12. On the other hand, the use of DM might be preferable over DDM in some cases due to the significantly smaller micelle size of DM [67].

Biophysical characteristics of the detergents used in the present study are shown in Appendix A. Due to large differences in CMC values and costs, detergents are not used at the same relative CMC value. This is also apparent from Appendix A, which details that detergents were used at concentrations ranging from 0.4× CMC to 17,667× CMC. An important issue that is usually not addressed in practical membrane protein purification is the dimension of each micelle. A membrane-spanning α-helix consists of approximately 20 amino acid residues since the hydrophobic part of a eukaryotic membrane is 30 Å. As is apparent from Appendix A, the tested detergents have a radius of gyration that matches this value.

Here, we used the knowledge gained from the initial screening phase to scale up the expression and purification of His-FICD (solubilised in DDM), resulting in the purification of 1–2 mg of protein per litre of yeast culture (Figure 5A–C). As alternatives to detergents, it may be worth pursuing styrene-maleic acid copolymers (SMALPs) for solubilisation [68] and insertion of the detergent solubilised FICD protein into phospholipid nanodiscs [69] to stabilise the protein. These approaches have the advantage that the membrane protein is maintained in a native lipid environment or at least a phospholipid environment.

Solubilised His-FICD adopted a single oligomeric state as judged by SEC analysis (Figure 5B), as is the case for N-terminally truncated FICD, which forms a highly stable homodimer in the solution and in all published structures, e.g. [10,11,12]. To assess the oligomeric state of His-FICD experimentally, we employed native mass spectrometry, but were unfortunately not able to detect the protein by this method (Drs. C. Sahin and M. Landreh, Karolinska Institutet, Stockholm, Sweden, personal communication). Still, based on the strictly dimeric behaviour of the protein produced in *E. coli*, we find it likely that His-FICD is also a dimer.

Similar to dimeric, N-terminally truncated FICD, full-length His-FICD readily deAMPylated its natural substrate BiP in vitro (Figure 5D,E). The kinetics of BiP deAMPylation displayed by His-FICD were similar to published data for BiP deAMPylation by N-terminally truncated FICD [9], with both showing deAMPylation half-times of 10–15 min with around 80% deAMPylation within 60 min, when conducting experiments under comparable conditions. We conclude that the conditions chosen for detergent solubilisation of His-FICD are well suited to preserve the structural integrity and function of the protein.

The expression platform developed in this study allowed the production of detergent-solubilised human FICD displaying deAMPylation activity. To our knowledge, this is the first time that FICD purified to homogeneity has been investigated in vitro in its full-length form. Of note, the transmembrane domain and juxta-membrane regions of single-pass membrane proteins are increasingly being recognised as functionally important, modulating for example protein–protein interactions, signalling and cellular localisation [70,71,72,73,74]. Investigating the previously uncharacterised N-terminal region of FICD may thus reveal the presence of unidentified structural motifs, modulator binding sites or other regulatory elements that would be impossible to identify when using a truncated variant of the enzyme. Our work provides a straightforward method for the recombinant production of single-pass transmembrane proteins to allow biochemical and biophysical investigation of the full-length proteins, and will help elucidate the structure and function of full-length FICD.

## 4. Materials and Methods

### 4.1. Yeast Strains

Expression of His_8_-GFP-TEV-FICD (GFP-FICD) and His_8_-FICD (His-FICD) was performed in the *S. cerevisiae* strain PAP1500 (*MATα ura3-52 trp1::GAL10-GAL4 lys2-801 leu2**Δ**1 his3**Δ**200 pep4::HIS3 prb1**Δ**1.6R can1 GAL*) [34].

### 4.2. Construction of S. cerevisiae Expression Plasmids

A cDNA encoding human FICD (Integrated DNA Technologies, Coralville, IA, USA; Gene ID: 11153), codon-optimised for expression in *S. cerevisiae,* was PCR amplified using Phusion DNA polymerase (Thermo Fisher Scientific, Waltham, MA, USA)) and the primers detailed in Appendix A to generate two constructs: FICD N-terminally tagged with either His_8_-yEGFP [75], including a TEV protease cleavage site (His_8_-GFP-TEV-FICD), or the His_8_ tag alone (His_8_-FICD). The plasmids were generated by in vivo homologous recombination in *S. cerevisiae* by transforming the PAP1500 strain with the appropriate PCR fragments and *Bam*HI/*Hin*dIII-digested pEMBLyex4 vector [35,44] (Figure 3A) according to [76]. PAP1500 transformants were selected on synthetic defined medium plates supplemented with 30 mg/L leucine and 20 mg/L lysine [34,77].

### 4.3. Protein Expression in S. cerevisiae

A single colony of transformed yeast cells was propagated initially in synthetic defined medium supplemented with 30 mg/L leucine and 20 mg/L lysine, and then in the same medium supplemented only with lysine [34]. For small-scale expression of His_8_-GFP-TEV-FICD, a leucine-starved culture was used to inoculate 2 L yeast extract-peptone-glycerol medium supplemented with 0.5% (*w*/*v*) glucose. The cultivation was continued at 25 °C with shaking until OD_450_ reached 1.0. Cultures were thermo-equilibrated to 15 °C and protein expression was induced by adding galactose to a final concentration of 2% (*w*/*v*). The large-scale expression of His_8_-FICD was carried out in 15 L Applikon Bioreactors (Getinge, Göteborg, Sweden) equipped with ADI 1030 Bio Controller as described previously [34]. Briefly, 1 L of leucine-starved preculture was used to inoculate 10 L of minimal medium supplemented with 3% glucose (*w*/*v*), 3% glycerol (*w*/*v*) and all amino acids except leucine and isoleucine. During growth the temperature was maintained at 20 °C, and the pH at 6.0 using automated titration with 20% NH_4_OH. When the glucose was used up, identified as the time point where the rate of NH_4_OH addition started to level off, the bioreactor was supplemented with an additional 2% (*w*/*v*) glucose. At the time when the rate of NH_4_OH addition started to level off again, the bioreactor was cooled to 15 °C and protein production was induced by the addition of galactose to a final concentration of 2% (*w*/*v*). The cells were harvested 120 h after induction by centrifugation at 1500× *g* for 10 min at 4 °C. The cells were washed with 0.9% (*w*/*v*) NaCl (His_8_-GFP-TEV-FICD) or 1 M glycerol (His_8_-FICD) to ensure isosmotic conditions, and the cell pellets were stored at −80 °C until use.

### 4.4. Isolation of Yeast Membranes

Crude yeast membranes were prepared by a glass bead disruption method [78]. Briefly, His_8_-GFP-TEV-FICD cell pellets were resuspended 1:3 (*w*/*v*) in ice-cold Lysis Buffer 1 (25 mM imidazole-HCl pH 7.5, 1 M NaCl, 10% glycerol (*w*/*v*), 1 mM EDTA, 1 mM EGTA, and an in-house protease inhibitor cocktail (1 mM PMSF, 1 μg/mL leupeptin, 1 μg/mL pepstatin and 1 μg/mL chymostatin)). The lysate was homogenised in a total volume of 30 mL including glass beads in a 50-mL tube by vortexing 8 times for 1 min. Unbroken cells and cell debris were removed by centrifugation at 1000× *g* for 10 min at 4 °C. Membranes were collected from the resulting supernatant by centrifugation at 160,000× *g* for 1 h at 4 °C. Membrane pellets were resuspended in Lysis Buffer 1, and stored at −80 °C until use. For large-scale preparation of His_8_-FICD-containing membranes, the cell pellets were resuspended 1:5 (*w*/*v*) in ice-cold Lysis Buffer 2 (40 mM Tris-HCl pH 8.0, 1 M KCl, 10% glycerol (*w*/*v*), 1 mM EDTA, 1 mM EGTA, 5 mM β-mercaptoethanol, and protease inhibitor cocktail), and the lysis was performed in a volume of 25 mL.

### 4.5. Detergent Screen

Membranes purified from yeast expressing His_8_-GFP-TEV-FICD were diluted to 2 mg/mL in Lysis Buffer 1 and supplemented with a detergent (see Table 1) at a final concentration of 2, 4 or 6 mg/mL. When indicated, CHS was included to one third of the detergent concentration. The mixtures were incubated at 4 °C for 1 h with gentle agitation. Insoluble material was collected by centrifugation at 160,000× *g* for 30 min at 4 °C. The solubilisation efficiency was estimated by measuring the GFP fluorescence (Fluoroskan Ascent, Thermo Fisher Scientific, Waltham, MA, USA; excitation at 485 nm and emission at 520 nm) in the soluble fraction before and after the centrifugation [32]. For FSEC, 350 μL of the solubilised membrane proteins were separated on a Superose 6 Increase 10/300 GL (Cytiva, Marlborough, MA, USA) column connected to an ÄKTA purifier (Cytiva, Marlborough, MA, USA) in SEC Buffer 1 (25 mM Tris-HCl pH 7.6, 500 mM NaCl, 10% glycerol, 0.3 mg/mL DDM). An in-line fluorescence detector (Shimadzu RF-20A prominence, Shimadzu Corporation, Kyoto, Japan) was used to record the elution of the GFP-tagged FICD. All detergents used for detergent screening were purchased from Affymetrix (Santa Clara, CA, USA) or GlyconBiochemicals (Luckenwalde, Germany) and are listed in Table 1.

### 4.6. Large-Scale Purification of His_8_-FICD

Membranes containing His_8_-FICD were diluted to a concentration of 5 mg/mL in Lysis Buffer 2 supplemented with 10 mg/mL DDM and 3.3 mg/mL CHS, and incubated overnight at 4 °C with gentle agitation. The supernatant was then circulated approximately 10 times through a HisTrap FF 1 mL column (Cytiva, Marlborough, MA, USA) connected to an ÄKTA Explorer (Cytiva, Marlborough, MA, USA). The purification was performed in IMAC Buffer (40 mM Tris-HCl pH 8.0, 500 mM KCl, 10% glycerol (*w*/*v*), 0.3 mg/mL DDM, 1 mM PMSF, and 1 mM β-mercaptoethanol), supplemented with imidazole as indicated below. The bound proteins were washed with 90 column volumes (CVs) of IMAC Buffer with 10 mM imidazole, and 45 CVs with 25 mM imidazole. The bound proteins were subsequently eluted with an imidazole gradient from 25 to 350 mM over 30 CV, followed by a 12 CV wash step with 500 mM imidazole. To check that His_8_-FICD was uniformly folded, 500 μL of IMAC-purified protein was loaded onto an Enrich SEC650 column (BioRad, Hercules, CA, USA) pre-equilibrated with SEC buffer 2 (40 mM Tris-HCl pH 8.0, 500 mM KCl, 10% glycerol (*w*/*v*), 0.3 mg/mL DDM, 1 mM PMSF, and 1 mM β-mercaptoethanol) and eluted isocratically over 1.5 CV. The purified protein was stored in the SEC Buffer 2 at 4 °C.

### 4.7. Whole-Cell Fluorescence and Live Cell Bioimaging

Whole-cell fluorescence was measured in 1 OD_450_ unit of cells resuspended in 200 μL of 0.9% (*w*/*v*) NaCl using a Fluoroskan Ascent microplate reader (Thermo Fisher Scientific, Waltham, MA, USA) with 485 nm excitation and 520 nm emission wavelengths. For bioimaging, 100 μL of *S. cerevisiae* expressing His_8_-GFP-TEV-FICD were harvested, washed in 0.9% NaCl and resuspended in 5 μL of 0.9% NaCl. In total, 1 μL was pipetted onto a glass slide (Delta Lab, Barcelona, Spain) and whole-cell GFP fluorescence was imaged at 1000 times magnification using a Nikon Eclipse E600 microscope (Nikon, Tokyo, Japan) coupled to an Optronics Magnafire model S99802 camera (Hensoldt, Taufkirchen, Germany). GFP-tagged versions of the endogenous Pmr1 [51] and Pmc1 [52] proteins were used to assess the localisation of recombinant GFP-tagged FICD. The localisation of Pmc1-GFP and Pmr1-GFP was established previously [35,50].

### 4.8. Quantification of His_8_-GFP-TEV-FICD Expression

The total protein concentration in the crude membranes was determined by the Lowry assay [79] as described previously [32]. To estimate the amount of FICD in the crude membranes, the GFP fluorescence was converted to pmol/mg membrane protein using a standard curve generated from purified GFP mixed with yeast membranes as previously established [32,35]. The fluorescence intensity was measured in the crude membranes, and converted into pmol/mg crude membranes using the equation:pmol = 0.0875 × RFU,(1)
where RFU denotes relative fluorescence units.

### 4.9. SDS-PAGE, in-Gel Fluorescence, and Immunoblotting

Proteins were separated on 10% SDS-PAGE gels and visualised by Coomassie staining or in-gel fluorescence on a ChemiDoc MP imaging system (Bio-Rad, Hercules, CA, USA) using excitation at 492 nm and emission at 510 nm. For immunoblotting, the proteins were transferred onto a polyvinylidene fluoride membrane (Immobilon-P, Merck Millipore, Darmstadt, Germany) in a Mini Trans-Blot (Bio-Rad) transfer system. A polyclonal rabbit antiserum (BioGenes, Berlin, Germany) raised against the entire ER-luminal region (residues 45–458) of human FICD produced in *E. coli* was used in combination with a horseradish peroxidase-conjugated α-rabbit secondary antibody (Pierce, Thermo Fisher Scientific, Waltham, MA, USA). The primary antibody was diluted 1:1000 in 2% skim milk dissolved in Tris-buffered saline with 0.1% Tween^®^ 20 detergent (TBST) before incubation of the membrane overnight at 4 °C. Secondary antibody was diluted 1:100,000 in 2% skim milk-TBST and incubated with the membrane for 50 min at room temperature before thorough washing with TBST and a final wash with TBS. Chemiluminescence detection was performed using ECL Select Peroxide and Luminol solutions (Cytiva, Marlborough, MA, USA) according to the manufacturer’s directions.

### 4.10. Construction of E. coli Expression Plasmids

The expression vector pET39_Ub19-His_10_-FICD_Δ102_ L258D (pLE932), encoding monomeric, soluble human FICD L258D, N-terminally tagged with ubiquitin (Ub) containing 10 consecutive histidine residues inserted into a loop region (Ub-His_10_), was generated from an Ellgaard lab cDNA encoding the entire ER-luminal region of FICD L258D (residues 103–458). The target region was amplified using Phusion DNA polymerase (Thermo Fisher Scientific, Waltham, MA, USA) and primers containing *Nco*I and *Bam*HI restriction sites (see Appendix A). The insert was subsequently ligated into the pET39_Ub19-His_10_ vector [80]. The His_6_-BiP_Δ18_ T229A V461F (pLE924) expression vector encoding human BiP without the signal sequence was generated by site-directed mutagenesis using a template vector pProEx Htc-His_6_-BiP_Δ18_ (a kind gift from Prof. Johannes Buchner, Technische Universität München) and the primers listed in Appendix A. The His_6_-BiP_Δ18_ T229A V461F (pLE924) double mutant is deficient in ATP hydrolysis (T229A) [81,82,83,84] and oligomerization (V461F) [85,86,87].

### 4.11. Expression and Lysis of Soluble Proteins in E. coli

For expression of Ub–His_10_-FICD_Δ102_ L258D and His_6_-BiP_Δ18_ T229A V461F, the pLE932 and pLE924 plasmids were transformed into BL21 (DE3) cells (Novagen, Merck, Darmstadt, Germany) and cultivated in Lysogeny Broth (LB) medium containing 50 μg/mL kanamycin (pLE932) or 100 μg/mL ampicillin (pLE924) at 37 °C to an OD_600_ of 0.8. The cultures were cooled to 4 °C before induction of protein expression with 0.5 mM isopropyl β-D-1-thiogalactopyranoside (IPTG). The expression was conducted for 18 h at 16 °C, before the cells were harvested by centrifugation at 6000× *g* for 15 min. The cell pellets were resuspended in 1:10 (*w*:*v*) Lysis Buffer 3 (50 mM NaH_2_PO_4_ pH 7.4, 300 mM NaCl, 10 mM imidazole, 5% glycerol) supplemented with 1 mg/mL lysozyme, 1 mM PMSF, 1× Complete protease inhibitor cocktail (Roche, Basel, Switzerland) and 250 U Benzonase (Sigma-Aldrich, Merck, Darmstadt, Germany). The lysis was completed by sonication (UP200S, Hielscher, Teltow, Germany), 12 times for 14 s. The cell debris was collected by centrifugation at 15,000× *g* for 30 min at 4 °C.

### 4.12. Purification of Soluble Proteins

The soluble proteins were purified using an ÄKTA Start chromatography system (Cytiva, Marlborough, MA, USA) mounted with a 5 mL HiFliQ Ni-nitrilo triacetic acid column (ProteinArk, Sheffield, UK) pre-equilibrated with Lysis Buffer 3. The column was washed with 15 CV of Lysis Buffer 3 containing 25 mM imidazole, and eluted with an imidazole gradient from 25 to 400 mM over 20 CV. For His_6_-BiP_Δ18_ T229A V461F the histidine tag was cleaved using TEV protease and removed by affinity purification on a Ni-NTA column [88]. The purified proteins were dialysed into Storage Buffer (50 mM NaH_2_PO_4_ pH 7.4, 300 mM NaCl) and stored at 4 °C.

### 4.13. Preparation of AMPylated BiP

Purified BiP was AMPylated with radioactively labelled α-[^32^P]-ATP essentially as described previously [9]. Briefly, 6 µg BiP_Δ18_ T229A V461F was AMPylated using 2 µM Ub–His_10_-FICD_Δ102_ L258D in AMPylation Buffer (25 mM HEPES-KOH, pH 7.4, 100 mM KCl, 1 mM MnCl_2_) containing 2 mM DTT, 15 µM ATP and 100 µCi α-[^32^P]-ATP in a 40 µL reaction volume and incubated at room temperature for 10 min. Then, another 12 µg BiP_Δ18_ T229A V461F was added and the reaction mixture was incubated for an additional 50 min. To AMPylate BiP_Δ18_ T229A V461F non-radioactively, 180 µg of the protein was AMPylated with 2.5 µg Ub–His_10_-FICD_Δ102_ L258D for 4 h at 30 °C in 2×AMPylation Buffer, containing 3 mM ATP. The 180 µg non-radioactive AMPylated BiP_Δ18_ T229A V461F and the 18 µg radioactively AMPylated BiP_Δ18_ T229A V461F were mixed and added with 1.2 mL high salt AMPylation buffer (25 mM HEPES-KOH, pH 7.4, 500 mM KCl, 2 mM MnCl_2_) before IMAC purification on Ni-NTA superflow resin (Qiagen, Hilden, Germany) using gravity flow, with the aim of removing Ub-His_10_-FICD_Δ102_ L258D from the mixture. Finally, the flow-through was concentrated on an Amicon Ultra-0.5 spin column (Sigma Aldrich), while exchanging the buffer into 2×AMPylation Buffer. This step removed the vast majority of ATP in the solution in seeking to avoid a competing AMPylation reaction in the following deAMPylation assay.

### 4.14. DeAMPylation Assay

To investigate His_8_-FICD deAMPylation activity, 2.8 µg AMPylated BiP_Δ18_ T229A V461F (a mixture of 2.52 µg non-radioactively AMPylated BiP and 0.28 µg radioactively AMPylated BiP) was incubated in deAMPylation Buffer (25 mM HEPES-KOH pH 7.4, 100 mM KCl, 5 mM MnCl_2_, 2 mM DTT, 0.2 mg/mL DDM) in the presence or absence of 0.26 µg His_8_-FICD in a final reaction volume of 15 µL. The negative control was used to ensure that the loss of AMP signal was not due to spontaneous hydrolysis or deAMPylation by Ub–His_10_-FICD_Δ102_ L258D [10]. Samples were incubated at room temperature and quenched at specified time points by boiling in reducing SDS-PAGE loading buffer. Proteins were separated by SDS-PAGE and analysed by Coomassie staining and phosphorimaging using a Typhoon FLA 700 phosphorimager (Cytiva, Marlborough, MA, USA). Protein band intensities were quantified using Fiji [89]. The radioactivity signal was related to the Coomassie signal, and the resulting values were normalised using the equation:y_norm_ = (y − y_min_)/(y_max_ − y_min_),(2)
where y_norm_ is the normalised value, y_max_ the maximum value of the dataset and y_min_ is assumed to be 0.

The standard error of the mean (SEM) was calculated using the equation:(3)SEM=SDn−1
where SD is the standard deviation and n is the number of experiments.

## Figures and Tables

**Figure 1 ijms-23-02458-f001:**
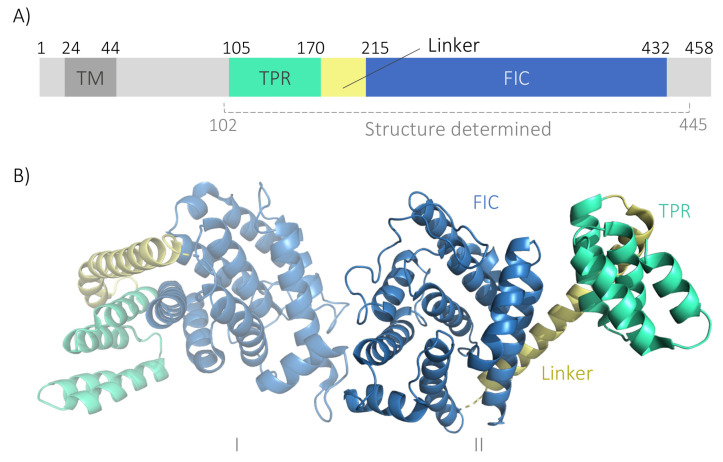
Schematic overview of the human FICD structure. (**A**) FICD is comprised of a short N-terminal cytosolic tail, a transmembrane domain (TM; dark grey), two tetratricopeptide repeats (TPR; turquoise), a linker (yellow), and the catalytic Fic domain (FIC; blue). The segment for which crystal structures have been solved (approximately residues 102 to 445 depending on the particular study) is indicated [10,11,12,22]. The numbers denote residue positions that mark the boundaries of key regions in the structure of FICD. (**B**) Crystal structure of the N-terminally truncated, soluble FICD_103–433_ dimer (PDB ID: 4U04) displays a helical conformation in which FICD forms an asymmetric, non-covalent dimer via the protomer Fic domains (I and II). Same colour scheme as Panel A. Created using PyMol [23].

**Figure 2 ijms-23-02458-f002:**
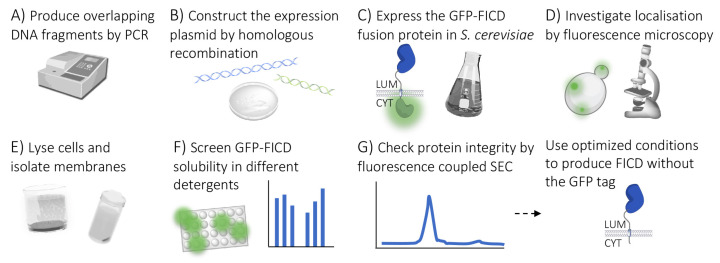
Workflow for screening GFP-FICD expression and purification conditions. (**A**,**B**) The expression vector was constructed from overlapping DNA fragments using homologous recombination in *S. cerevisiae*. (**C**) PAP1500 expression system was used to express GFP-tagged FICD, (**D**) and membrane insertion was checked by fluorescence microscopy. (**E**) Cells were subsequently homogenised and the membranes isolated. (**F**) GFP fluorescence was used to quantify solubilisation efficiency and identify the best detergents. (**G**) Homogeneity of solubilised FICD was determined using fluorescence-coupled size exclusion chromatography (FSEC). LUM; ER lumen, CYT; cytosol.

**Figure 3 ijms-23-02458-f003:**
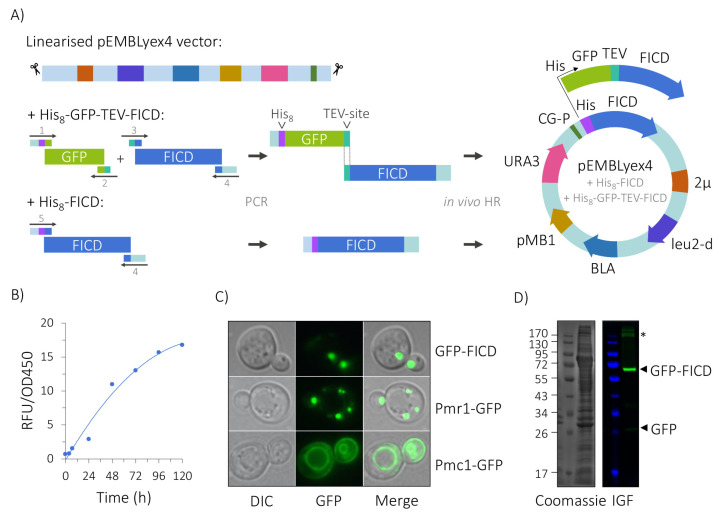
Generation and expression of FICD constructs. (**A**) FICD was tagged with His_8_-GFP-TEV (where TEV denotes a tobacco etch virus protease cleavage site) or His_8_ alone and ligated into linearised pEMBLyex4 vector by in vivo homologous recombination (HR) in *S. cerevisiae*. The His_8_-GFP-TEV, TEV-FICD and His_8_-FICD DNA fragments were constructed by PCR using appropriate cDNA templates and primers containing overlapping sequences (see Appendix A for primer details). CG-P, CYC-GAL promoter; *leu2-d*, a poorly expressing allele of the β-isopropylmalate dehydrogenase gene; *URA3*, selective marker gene encoding the orotidine-5′-phosphate decarboxylase enzyme; *BLA*, β-lactamase gene; 2μ, 2-μm origin of replication; pMB1, bacterial origin of replication. (**B**) Time dependent accumulation of His_8_-GFP-TEV-FICD (GFP-FICD) after galactose induction at 15 °C, with whole-cell fluorescence measured in relative fluorescence units (RFU) in 1 OD_450_ unit of cells. (**C**) Cellular localisation of GFP-FICD, Pmr1-GFP (Calcium-transporting ATPase 1 (Golgi)) and Pmc1-GFP (Calcium-transporting ATPase 2 (vacuole)), imaged by fluorescence and phase contrast microscopy. (1000×). (**D**) SDS-PAGE analysis of GFP-FICD-containing membranes (corresponding to 200 RFU) by Coomassie staining (left) and in-gel fluorescence (IGF, right). High molecular weight species are marked with an asterisk (*).

**Figure 4 ijms-23-02458-f004:**
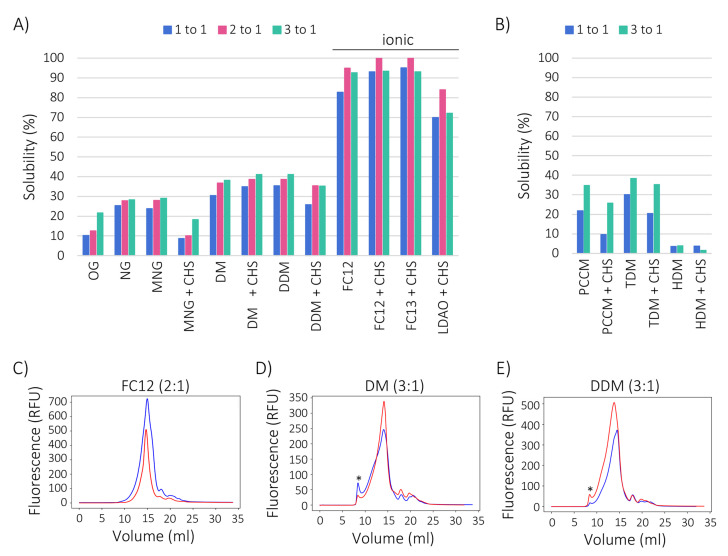
GFP-FICD maintains a folded conformation upon solubilisation. (**A**) Percentage solubility of GFP-FICD in the indicated detergent conditions (see Table 1 for detergent names and properties). When noted, solubilisation was performed in the presence of CHS. Fluorescence intensity was measured in the soluble fraction after ultracentrifugation, and solubility was quantified as the proportion of fluorescence in the soluble fraction relative to total membrane fluorescence. Solubility in three different detergent to protein ratios (1:1, 2:1 and 3:1) is shown. (**B**) GFP-FICD solubility measured as in A, but in two detergent to protein ratios (1:1 and 3:1). (**C**–**E**) FSEC profiles of GFP-FICD solubilised in the indicated detergents in the highest yielding detergent to protein ratio (FC12; 2:1 (**C**), DM; 3:1 (**D**), DDM; 3:1 (**E**)). An in-line fluorescence detector was used to record the elution profile of GFP-FICD. Blue line, without CHS; red line, with CHS. Fluorescent material eluting in the void volume is denoted by an asterisk (*).

**Figure 5 ijms-23-02458-f005:**
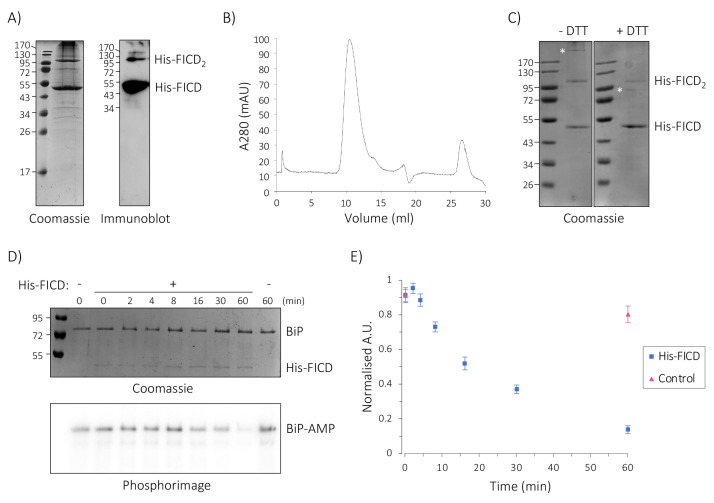
Solubilised His-FICD can be purified in a correctly folded, active conformation. (**A**) Non-reducing SDS-PAGE gel stained with Coomassie (left) and an immunoblot developed with α-FICD antiserum (right) of the IMAC purified FICD. Monomeric His-FICD (53 kDa) and dimeric His-FICD_2_ (106 kDa) are annotated. (**B**) SEC analysis of IMAC-purified FICD. Protein absorbance (A_280_) is plotted against the elution volume. (**C**) Non-reducing (left) and reducing (right) SDS-PAGE gel analysis of SEC-purified FICD. Where indicated, proteins were reduced using 10 mM dithiothreitol (DTT). Unidentified bands are marked with an asterisk (*). (**D**) Time-dependent deAMPylation of radioactively labelled BiP-α-[^32^P]-AMP in the presence and absence of His-FICD. The reaction was sampled and quenched at indicated time-points and analysed by SDS-PAGE. Coomassie stain (top) depicts the total protein content of the reaction, and the phosphorimage (bottom) shows the radioactive BiP-AMP signal. Representative gel and phosphorimage for seven experiments. (**E**) Quantification of deAMPylation of BiP-AMP in the presence (blue squares) and absence (red triangles) of His-FICD. Plotted are normalised mean values ± standard error of the mean (SEM) from seven replicates, performed in four independent experiments (one single experiment and three performed in duplicate).

**Table 1 ijms-23-02458-t001:** Detergents used in the screening phase.

Abbreviation	Detergent Name	Properties
MNG	2,2-Dioctylpropane-1,3-bis ß-D-maltopyranoside	Non-ionic
DM	n-decyl ß-D-maltopyranoside	″
DDM	n-dodecyl ß-D-maltopyranoside	″
TDM	n-tridecyl ß-D-maltopyranoside	″
HDM	n-hexadecyl ß-D-maltopyranoside	″
OG	n-octyl ß-D-glucopyranoside	″
NG	n-nonyl-β-D-glucopyranoside	″
PCCM	4-trans-(4-trans propylcyclohexyl)-cyclohexyl α-D-maltopyranoside	″
FC12	n-dodecylphosphocholine	Ionic
FC13	n-tridecylphosphocholine	″
LDAO	n-dodecyl-dimethylamine oxide	″
CHS	cholesteryl hemisuccinate	Ionic

## Data Availability

The data presented in this study are available on request from the corresponding author.

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
