# Peer review of "Production of an Active, Human Membrane Protein in *Saccharomyces cerevisiae*: Full-Length FICD"

_ijms, 2022, doi:10.3390/ijms23052458_

Round 1
Reviewer 1 Report
The first round of review (Reconsider after major revision)
The manuscript is well written, experiments are well executed and the data is presented in an organized manner.
The following concerns could be addressed.
- The detergent screening is done merely on the basis of extraction efficiency and the oligomerization profile on FSEC, Could the detergent screen be repeated based on the activity assay? The detergent which keeps the protein in the best-folded form should be able to retain the highest activity. The extraction efficiency alone does not point towards the best-suited detergent for this system. The soluble ectodomain of FICD could be used as a control for an activity-based selection of detergents.
- Does the choice of detergent have an effect on the oligomerization state?
- The figures 5F and S1B show the deAMPylation activity in a rather qualitative way. Could this be compared with the reported activity range in the presence of soluble FICD? The fact that the protein is active alone does not inform much about the right fold of the protein. This could very well be a small fraction of the full activity. Hence a quantification of the activity (specific activity if possible) is crucial to ensure that the authors have chosen the best conditions for the membrane protein.
- Optimizing the conditions in the presence of a large solubilization tag such as GFP and using those optimized conditions to express and purify untagged FICD is not an ideal strategy since GFP (28 kDa) will have a big effect on the solubility of the fusion protein. The GFP-tagged FICD and the His-FICD could have very different physical properties.
The second round of review
The authors addressed all the queries from the last round of the review process with additional experiments, which satisfy the concerns. Some minor comments could be addressed.
a) Table 1 and 3 could be moved to supplementary material in the interest of space.
b) It will be informative to add critical micellar concentrations (cmc), the percentage /concentration of the detergents used, micellar size, in the experiments used vs the previously reported concentration used, etc. in Table 2. Any correlations between this and the size of the transmembrane helix (20 aa in this case) could be discussed in the discussion section.
c) Figure 5A needs some formatting, this chromatogram could also be skipped since Ni-NTA is very routine and the chromatogram might not provide any indication about the quality of the protein, unlike the SEC chromatogram.
d) The authors could mention alternative strategies such as styrene-maleic acid copolymers or phospholipid nanodiscs.
Author Response
The first round of response
We thank both reviewers for their constructive and insightful feedback, which has clearly helped improve our manuscript. Below, we provide a detailed point-by-point response to the questions raised for the original submission:
Overall comments from the reviewer:
The manuscript is well written, experiments are well executed and the data is presented in an organized manner.
The following concerns could be addressed:
- The detergent screening is done merely on the basis of extraction efficiency and the oligomerization profile on FSEC. Could the detergent screen be repeated based on the activity assay? The detergent which keeps the protein in the best-folded form should be able to retain the highest activity. The extraction efficiency alone does not point towards the best-suited detergent for this system. The soluble ectodomain of FICD could be used as a control for an activity-based selection of detergents.
Answer:
As pointed out by the reviewer, the extraction efficiency alone does not necessarily identify the best-suited detergent. Therefore, we based our decision about the detergent conditions on other additional considerations, i.e. a monodisperse SEC profile as well as compatibility with structural studies. With our newly performed deAMPylation assays (see Point #3 below), we were able to demonstrate that using the chosen detergent conditions, full-length FICD shows deAMPylation activity that is comparable to what has previously been determined for the dimeric N-terminally truncated protein produced in E. coli (ref. 9 in the manuscript).
- Does the choice of detergent have an effect on the oligomerization state?
Answer:
While we tried to determine the oligomerization state of full-length FICD using native MS, this method did unfortunately not produce positive results, as detailed in our answer to Reviewer 1, Point #2 (please see above). We do therefore currently not have a method to determine the exact oligomeric state of FICD produced in our system, but find it likely that it forms a dimer (see Point #2, Reviewer 1). On this background, we can unfortunately not provide a direct answer to the question. However, during detergent screening we observed that when using cholesteryl hemisuccinate (CHS) in combination with the n-tridecyl ß-D-maltopyranoside (TDM) and 4-trans-(4-trans propylcyclohexyl)-cyclohexyl α-D-maltopyranoside (PCCM), the FSEC elution profiles displayed a “shoulder” (at higher elution volumes) on the peak representing FICD, potentially indicating a partial dimer dissociation for these particular combinations of detergents.
3.The figures 5F and S1B show the deAMPylation activity in a rather qualitative way. Could this be compared with the reported activity range in the presence of soluble FICD? The fact that the protein is active alone does not inform much about the right fold of the protein. This could very well be a small fraction of the full activity. Hence a quantification of the activity (specific activity if possible) is crucial to ensure that the authors have chosen the best conditions for the membrane protein.
Answer:
We thank the reviewer for the suggestion to improve the deAMPylation assay in Fig. 5E-F. We have now done so by performing four independent assays with newly produced full-length FICD (Fig. 5E-F).
While it is true that the observed activity in this assay could potentially be caused by a small fraction of correctly folded protein and that the majority of polypeptide chains could adopt a non-native (inactive) conformation, based on the monodisperse SEC profile we do not find this very likely. In support, our new experiments demonstrated that His-FICD readily removed ~80% of the AMP from BiP within 60 minutes with a half-time for deAMPylation of approximately 15 minutes (Fig. 5F).
A similar experiment, using N-terminally truncated recombinant FICD produced in E. coli has previously been published in ref. 9, Fig. 2A:
The two experiments were conducted using similar experimental conditions in terms of temperature, enzyme to substrate ratio and protein concentrations. Although the deAMPylation kinetics in the experiment shown above may be slightly faster (no quantification of this experiment was performed in the publication) than in our own experiment, the kinetics is similar. Overall, we are therefore confident that the conditions chosen are well suited to preserve the structural integrity and function of the protein.
- Optimizing the conditions in the presence of a large solubilization tag such as GFP and using those optimized conditions to express and purify untagged FICD is not an ideal strategy since GFP (28 kDa) will have a big effect on the solubility of the fusion protein. The GFP-tagged FICD and the His-FICD could have very different physical properties.
Answer:
Optimizing production of recombinant membrane proteins, and subsequently solubilization and purification protocols, are essential in order to obtain prime quality membrane proteins for downstream applications, including structure determination and biochemical characterization. In this regard, the tagging of the protein of interest with GFP in order to increase the efficiency of these processes has turned out the be a highly successful approach. So, the approach we have taken is well described in the literature by us and others, and although tagging a protein with GFP may certainly affect solubilization efficiency and accumulation of a recombinant membrane protein, it has turned out to be a very operational approach that has resulted in a large number of high-resolution structures (exemplified by the four references given below).
- Yiming Niu, Xiao Tao, Kouki K Touhara, and Roderick MacKinnon (2020) Cryo-EM analysis of PIP2regulation in mammalian GIRK channels. eLife 2020; 9: e60552.
- Ji Sun and Roderick MacKinnon (2020) Structural Basis of Human KCNQ1 Modulation and Gating. Cell 180:2, 340-347.
- Kaituo Wang, Sarah Spruce Preisler, Liying Zhang, Yanxiang Cui, Julie Winkel Missel, Christina Grønberg, Kirstine Callø, Pascal F. Egea, Dan Arne Klærke, Michael Pusch, Per Amstrup Pedersen, Z. Hong Zhou & Pontus Gourdon (2019) Structure of the human ClC-1 chloride channel. PLoS Biol. 25;17(4):e3000218.
- Kamil Gotfryd, Andreia Filipa Mósca, Julie Winkel Missel, Sigurd Friis Truelsen, Kaituo Wang, Mariana Spulber, Simon Krabbe, Claus Hélix-Nielsen, Umberto Laforenza, Graça Soveral, Per Amstrup Pedersen and Pontus Gourdon (2018). Human adipose glycerol flux is regulated by a pH gate in AQP10. Nat Commun. 9(1):4749.
The second round of response
We thank the reviewer for these new comments that have helped improve the manuscript further.
Response to comments:
a) We have moved these two tables to the Supplementary Materials.
b) We have added a new section to the Discussion to discuss the points mentioned by the reviewer. In the interest of space, we have added a new table (Table S2) to the Supplementary Materials, which details CMC, concentrations used for solubilization, solubilization conditions and radius of gyration (rather than adding this information to the existing table in the main text).
c) We have deleted Fig. 5A and reformatted the figure.
d) As suggested, we have now added a few sentences to the Discussion concerning styrene-maleic acid copolymers or phospholipid nanodiscs.
Reviewer 2 Report
The first round of review (Accept after minor revision)
In this article, Virolainen and colleagues present an efficient way to express and purify human membrane protein FICD. They demonstrate its activity in the presence of detergent (DDM and DM) and show strong evidence that the enzyme is purified as a dimer. As previous biochemical and biophysical studies were done on the soluble part of the enzyme overproduced in E.coli, the ability to obtain pure, folded and active full-length protein is certainly a worthwhile achievement. My only concern is that this research article reads more like a protocol, and while I am convinced that such protocol should be accessible to the scientific community by means of publication, I can’t help but wonder whether a protocol paper wouldn’t be better adapted. This is however editorial decision; science-wise, this paper is sound.
I only have minor comments.
lines 207 and 233: Data not shown. Please show the data, negative results are better out there.
Related to the previous comment, could the authors elaborate on why no autoAMPylation activity is observed? Is it full-length FICD always 100% dimeric, whatever the concentration?
Why are there no error bars for the activity assay? Is there a reason not to perform measurements in triplicates? Furthermore, could the authors show both datasets they acquired?
In the methods, please avoid sentences like “as described previously” (e.g. line 392) (although I guess it’s acceptable for the Lowry assay as it is standard). Instead, please detail the protocol.
Line 343: Expression of His8-TEV-GFP-FICD (GFP-FICD). TEV and GFP have been swapped.
Line 367: why wash with 1M glycerol?
The second round of review
I am happy with the revised version of the manuscript. This is publication-worthy.
Author Response
Response to reviewer:
The first round of response
We thank both reviewers for their constructive and insightful feedback, which has clearly helped improve our manuscript. Below, we provide a detailed point-by-point response to the questions raised for the original submission:
Overall comments from the reviewer:
In this article, Virolainen and colleagues present an efficient way to express and purify human membrane protein FICD. They demonstrate its activity in the presence of detergent (DDM and DM) and show strong evidence that the enzyme is purified as a dimer. As previous biochemical and biophysical studies were done on the soluble part of the enzyme overproduced in E.coli, the ability to obtain pure, folded and active full-length protein is certainly a worthwhile achievement. My only concern is that this research article reads more like a protocol, and while I am convinced that such protocol should be accessible to the scientific community by means of publication, I can’t help but wonder whether a protocol paper wouldn’t be better adapted. This is however editorial decision; science-wise, this paper is sound.
Minor reviewer comments:
- lines 207 and 233: Data not shown. Please show the data, negative results are better out there.
Answer:
We have now added a new Fig. S1 to show the inefficient removal of the GFP tag by TEV protease.
- Related to the previous comment, could the authors elaborate on why no autoAMPylation activity is observed? Is it full-length FICD always 100% dimeric, whatever the concentration?
Answer:
As mentioned above, we have deleted the brief statement concerning autoAMPylation.
Re. dimerization: It is well established that FICD catalyses two antagonistic reactions by utilising an oligomerisation-induced switch in the structure of the active site; AMPylation by the monomeric enzyme and deAMPylation by the dimeric enzyme. In vitro, it has been demonstrated that recombinant purified FICD devoid of the first approximately 100 residues is dimeric in solution with an equilibrium dissociation constant in the low nanomolar range (KD = 1.2 nM; ref. 9 in the manuscript).
Unfortunately, we have not been able to experimentally address if our recombinant full-length FICD is dimeric or monomeric. Various methods – such as SEC, chemical crosslinking and SEC-MALS – may in principle be able to determine the oligomeric state of full-length FICD. However, due to the presence of detergent/micelles none of these methods would be likely to provide a definite answer. Instead, we attempted to unequivocally demonstrate dimer formation of His-FICD using native mass spectrometry in a new collaboration set up during revision of the manuscript with the lab of Prof. Michael Landreh, Karolinska Institutet, Stockholm. However, despite testing a variety of conditions (e.g. different buffer and detergent combinations), we were not able to detect the protein in the spectra.
In conclusion, since N-terminally truncated FICD has been shown to exist exclusively as a dimer in solution (see e.g. ref. 9 and 12 in the manuscript) and since our full-length FICD efficiently deAMPylates BiP, we find it likely that the protein we employed in the deAMPylation assay also exists as a dimer. Compared to the original submission, we have explained this point in more detail in the Discussion of the revised manuscript.
- Why are there no error bars for the activity assay? Is there a reason not to perform measurements in triplicates? Furthermore, could the authors show both datasets they acquired?
Answer:
We thank the reviewer for the suggestion to improve the deAMPylation assay in Fig. 5E-F. We have now done so by performing four independent assays with newly produced full-length FICD (Fig. 5E-F). As explained in the manuscript and in the detailed answer to Reviewer 2, Point #3 (see below), the deAMPylation activity of full-length FICD in our assay is comparable to the activity published for dimeric N-terminally truncated FICD expressed in E. coli.
- In the methods, please avoid sentences like “as described previously” (e.g. line 392) (although I guess it’s acceptable for the Lowry assay as it is standard). Instead, please detail the protocol.
Answer:
This wording has been removed, and replaced with detailed protocols inserted in the Materials and Methods section. In one instance, the standard procedure for transformation of yeast cells, we have added a new reference to the original procedure rather than providing a detailed protocol.
- Line 343: Expression of His8-TEV-GFP-FICD (GFP-FICD). TEV and GFP have been swapped.
Answer:
This mistake has been corrected.
- Line 367: why wash with 1M glycerol?
Answer:
1 M glycerol is included to have isosmotic conditions (same osmolarity as inside the cells), as cells sometimes are osmotically challenged due to the recombinant membrane protein accumulation. This is now briefly mentioned in the text in Section 4.3.
The second round of response
We thank the reviewer for a constructive review process
This manuscript is a resubmission of an earlier submission. The following is a list of the peer review reports and author responses from that submission.
Round 1
Reviewer 1 Report
In this article, Virolainen and colleagues present an efficient way to express and purify human membrane protein FICD. They demonstrate its activity in the presence of detergent (DDM and DM) and show strong evidence that the enzyme is purified as a dimer. As previous biochemical and biophysical studies were done on the soluble part of the enzyme overproduced in E.coli, the ability to obtain pure, folded and active full-length protein is certainly a worthwhile achievement. My only concern is that this research article reads more like a protocol, and while I am convinced that such protocol should be accessible to the scientific community by means of publication, I can’t help but wonder whether a protocol paper wouldn’t be better adapted. This is however editorial decision; science-wise, this paper is sound.
I only have minor comments.
lines 207 and 233: Data not shown. Please show the data, negative results are better out there.
Related to the previous comment, could the authors elaborate on why no autoAMPylation activity is observed? Is it full-length FICD always 100% dimeric, whatever the concentration?
Why are there no error bars for the activity assay? Is there a reason not to perform measurements in triplicates? Furthermore, could the authors show both datasets they acquired?
In the methods, please avoid sentences like “as described previously” (e.g. line 392) (although I guess it’s acceptable for the Lowry assay as it is standard). Instead, please detail the protocol.
Line 343: Expression of His8-TEV-GFP-FICD (GFP-FICD). TEV and GFP have been swapped.
Line 367: why wash with 1M glycerol?
Reviewer 2 Report
The manuscript is well written, experiments are well executed and the data is presented in an organized manner.
The following concerns could be addressed.
- The detergent screening is done merely on the basis of extraction efficiency and the oligomerization profile on FSEC, Could the detergent screen be repeated based on the activity assay? The detergent which keeps the protein in the best-folded form should be able to retain the highest activity. The extraction efficiency alone does not point towards the best-suited detergent for this system. The soluble ectodomain of FICD could be used as a control for an activity-based selection of detergents.
- Does the choice of detergent have an effect on the oligomerization state?
- The figures 5F and S1B show the deAMPylation activity in a rather qualitative way. Could this be compared with the reported activity range in the presence of soluble FICD? The fact that the protein is active alone does not inform much about the right fold of the protein. This could very well be a small fraction of the full activity. Hence a quantification of the activity (specific activity if possible) is crucial to ensure that the authors have chosen the best conditions for the membrane protein.
- Optimizing the conditions in the presence of a large solubilization tag such as GFP and using those optimized conditions to express and purify untagged FICD is not an ideal strategy since GFP (28 kDa) will have a big effect on the solubility of the fusion protein. The GFP-tagged FICD and the His-FICD could have very different physical properties.